# Gnb5 is a negative regulator of the BACE1-mediated Aβ generation and ameliorates cognitive deficits in a mouse model of Alzheimer's disease

Shaokun Chen[1], Jiechao Zhou[1], Shuzhong Wang[1], Erqu Chen[1], Kai Zhuang[1], Raozhou Lin[2], Chensi Liang[1], Dan Can[1], Huifang Li[1], Jing Li[3], Jie Zhang[1,4,5]*

1 Fujian Provincial Key Laboratory of Neurodegenerative Disease and Aging Research, Institute of Neuroscience, School of Medicine, Xiamen University, Xiamen, Fujian, China, 2 Broad Institute of MIT and Harvard, Cambridge, Massachusetts, United States of America, 3 Department of Neurointervention, The First Affiliated Hospital of Zhengzhou University, Henan Provincial Neurointerventional Engineering Research Center, Zhengzhou, Henan, China, 4 The Key Laboratory of Neural and Vascular Biology, Ministry of Education, Hebei Medical University, Shijiazhuang, Hebei, China, 5 Institute of Neuroscience, Fujian Medical University, Fuzhou, Fujian, China

* jiezhang@xmu.edu.cn

## Abstract

β-Amyloid (Aβ) is generated from the amyloid precursor protein (APP) through sequential cleavage by β-site APP-cleaving enzyme 1 (BACE1) and γ-secretase, where BACE1 acting as the rate-limiting enzyme. Elevated BACE1 levels in the brains of Alzheimer's disease (AD) patients implicate that dysregulated BACE1 expression is crucial to AD pathogenesis. However, the underlying regulatory mechanisms remain unclear. Here, we identified that the G protein subunit β5 gene (*Gnb5*), a component of the G protein-coupled receptor (GPCR) signaling pathway, is significantly downregulated in both human AD patients and AD mouse models. Conditional knockout of *Gnb5* in excitatory neurons resulted in cognitive impairments, whereas adeno-associated virus (AAV)-mediated overexpression of Gnb5 in the hippocampus ameliorated cognitive deficits and reduced Aβ deposition in 5xFAD mice. Mechanistically, we demonstrated that Gnb5 interacts with BACE1, modulating its expression and potentially influencing Aβ generation. We further identify the first tryptophan–aspartate domain (WD domain) of Gnb5 and the Ser81 residue as crucial for this regulation. Expression of this WD domain alone is sufficient to reduce Aβ deposition in 5xFAD mice, whereas a point mutation at Ser81 (S81L) abolishes this effect. Overall, our findings establish Gnb5 as a negative regulator of the BACE1-APP processing axis and unveil mechanistic insights into its role in Aβ-mediated AD pathogenesis.

**Data availability statement:** All relevant data are within the paper and its Supporting information files.

**Funding:** This work was supported by the National Natural Science Foundation of China (Grants: U23A20430, 81925010, 91849205, 92149303 to Jie Z); The Fundamental Research Funds for the Central Universities (Grants: 20720180049 and 20720190118 to Jie Z); The National Key Research and Development Program of China (Grant: 2021YFA1101402 to Jie Z); The Opening Foundation of Key Laboratory of Neural and Vascular Biology, Ministry of Education of China (Grant: NV20230010 to Jie Z). The funders had no role in study design, data collection and analysis, decision to publish, or preparation of the manuscript.

**Competing interests:** The authors have declared that no competing interests exist.

**Abbreviations:** AAV, adeno-associated virus; Aβ, β-amyloid; AD, Alzheimer's disease; APP, amyloid precursor protein; BACE1, β-site APP cleaving enzyme 1; CO-IP, co-immunoprecipitation; DEGs, differentially expressed genes; FRET, fluorescence resonance energy transfer; GPCR, G protein-coupled receptor; MWM, Morris water maze.

## Introduction

Alzheimer's disease (AD) is a neurodegenerative disorder characterized by progressive memory loss and cognitive impairment, representing the most common cause of dementia [1,2]. The pathological hallmarks of AD include the deposition of β-amyloid (Aβ) plaques and hyperphosphorylation of tau proteins leading to neurofibrillary tangles [3,4]. Accumulation of Aβ plaques is strongly associated with the synaptic dysfunction, neuronal death, and consequently cognitive decline [5]. Aβ is primarily generated from the cleavage of amyloid precursor protein (APP), with the initial step being the action of β-site APP-cleaving enzyme 1 (BACE1), which produces soluble APPβ (APPsβ) and the β-carboxy terminal fragment (β-CTF), and followed by γ-secretase-mediated cleavage to release Aβ [6]. Inhibiting BACE1 is being explored as a potential therapeutic approach for AD due to its role in the production of Aβ peptides [7]. Several BACE1 inhibitors have been developed and tested in preclinical and clinical trials. However, challenges such as side effects (including cognitive worsening in some cases) and dose-limiting toxicities have been encountered, leading to the discontinuation of some clinical trials [8–10]. Therefore, identifying novel targets to regulate BACE1 may have greater potential clinical value.

GPCRs signaling is mediated through the activation of heterotrimeric G proteins, comprising Gα, Gβ, and Gγ subunits. The Gβ subunit family consists of five members (Gnb1 to Gnb5), with Gnb5 being preferentially expressed in the brain and nervous system [11]. Unlike other Gβ, Gnb5 is regarded as a distinctive subunit due to its role in diverse intracellular signaling pathways interacts specifically with the R7 family of regulator of G protein signaling (RGS) proteins to form functional dimers [12,13]. Loss function of Gnb5 has been linked to neurological disorders such as epilepsy, retinal diseases, developmental disabilities, and hyperactivity [14–17]. Furthermore, the residue S81L mutations in Gnb5 gene have been linked to neuropsychiatric disorder and cognitive deficits, though how this mutation influences these outcomes is still largely unclear [18,19]. Recently, the variants of Gnb5 are reported to be associated with the onsets of Alzheimer disease, suggesting the potential roles of Gnb5 in AD [20]. However, the underlying mechanism in AD pathophysiology has yet to be clearly defined.

In this study, we demonstrate that Gnb5 regulates Aβ deposition and cognitive deficits by modulating BACE1 in AD. We found that Gnb5 is downregulated in both AD patients and mouse model. Overexpression of Gnb5 in the hippocampal of 5xFAD mice enhances cognitive function and decreases Aβ deposit. In contrast, excitatory neuron-specific knockout of Gnb5 results in severe cognitive deficits and increased Aβ plaque accumulation in 5xFAD mice. The mechanism involves Gnb5 binding to BACE1 and regulating its expression. The first WD repeat domain contains serine 81, which is essential for this interaction. Our research elucidates Gnb5 as a critical regulator of BACE1, offering valuable insights into Aβ pathogenesis in AD.

## Results

### Gnb5 is downregulated in the brains of AD patients and mouse models

G proteins are key signal transduction molecules implicated in AD pathogenesis, regulating Aβ generation and deposition [21,22]. To explore the potential effects of

G protein subunits in AD, the mRNA levels of all G protein subunits in the hippocampal and cortical tissues of 6-month-old 5xFAD mice and WT control mice were measured. This analysis revealed a consistent decrease in *Gnb5* expression in both regions (Fig 1A and 1B). Next, we analyzed four independent GEO datasets GSE1297 [23], GSE28146 [24], GSE36980 [25], and GSE48350 [26] to examine differentially expressed genes (DEGs) in hippocampal tissues of AD patients and controls. Using the GEO2R tool and re-analyzed the dataset [27,28], we identified 132 consistently DEGs, with *Gnb5* was among the most significantly down-regulated genes (*p*-value < 0.05; Fold-Change > 1.2) (Fig 1C). We then examined the ontology of the DEGs and identified 11 genes were linked to GPCR signaling. Notably, *Gnb5* was consistently downregulated in these AD human datasets (Fig 1D and 1E). To further characterize the association of *Gnb5* with late-onset Alzheimer's disease (LOAD), we analyzed a large dataset (GSE44772) [29] containing extensive clinical data from AD patients. Our analysis confirmed that *Gnb5* expression was reduced in the brain tissues of AD patients (Fig 1F). Additionally, *Gnb5* expression was negatively correlated with the age, Braak stage, and the degree of frontal atrophy (S1A–S1C Fig). Moreover, with similar reductions observed in others brain regions, including the entorhinal cortex, frontal cortex, and temporal cortex (S1D–S1F Fig).

We then confirmed that the expression of Gnb5 is significantly decreased in both the hippocampal and cortical of 3- and 6-months-old 5xFAD mice compared with their controls (Figs 1G–1J and S1G–S1J). Immunofluorescence staining also revealed a pronounced reduction in Gnb5 expression in AD brain compared with control (Fig 1K and 1L). Aβ oligomer (oAβ$_{1-42}$) administration also decreases the Gnb5 expression in primary cultured neurons at a time-dependent manner (S1K and S1L Fig). Thus, we found that the expression of Gnb5 is significantly downregulated in AD, which suggests its potential role in the onset and progression of AD.

### Excitatory neuron-specific deletion of *Gnb5* leads to cognitive impairments

Gnb5 has been reported to be predominantly expressed in the brain and nervous system [11,30]. Our tissue-wide analysis corroborated these findings, showing the highest expression levels in mouse brains (S2A Fig). We extended our analysis by profiling Gnb5 expression at both protein and mRNA levels in primary cultured neural cells, including neurons, astrocytes, and microglia. Quantitative analysis demonstrated a predominant neuronal expression of Gnb5, with significantly higher expression levels compared to glial cell populations (S2B–S2D Fig). In parallel, immunofluorescence co-staining of Gnb5 with cell-type-specific markers in mouse brain tissue sections revealed selective co-localization with neuronal markers (S2E Fig), further supporting its neuron-enriched expression pattern. Furthermore, we performed triple labeling of NeuN, Gnb5, and Thioflavin-S (Thio-S) in brain sections from both 5xFAD and wild-type mice. The 5xFAD group demonstrated a marked reduction in Gnb5 expression, particularly evident in the overall neuronal expression pattern, with a pronounced deficiency observed in regions containing Aβ plaques (S2F Fig).

Considering the predominantly neuronal expression of Gnb5, we generated *Gnb5* floxed mice (*Gnb5^{F/F}*) and crossed them with CamKIIα-Cre mice to achieve neuron-specific *Gnb5* knockout mice (Gnb5-CCKO) (Figs 2A, 2B and S2G–S2I). The knockout efficiency of *Gnb5* was confirmed by western blotting and qPCR in the hippocampus (Fig 2C and 2D). We did not observe any difference in the body weight of Gnb5-CCKO mice with control *Gnb5^{F/F}* mice, indicating that neuronal knockout of *Gnb5* does not cause developmental impairment (S2J Fig).

We subsequently conducted a series of behavioral assessments on Gnb5-CCKO and *Gnb5^{F/F}* mice at 3 months old. The open field and rotarod tests indicated that the locomotor abilities of Gnb5-CCKO mice remained intact (Figs 2E and S2K). Then, we focused on hippocampus-dependent learning and memory behavioral tasks, such as Morris water maze (MWM) and contextual fear conditioning test, which are widely used to assess spatial memory deficits in mouse models of AD [31]. During the six-day acquisition phase of the MWM training, Gnb5-CCKO mice exhibited pronounced spatial learning deficits, as evidenced by significantly longer latency to locate the hidden platform in the target quadrant compared to Gnb5^{F/F} littermates (Fig 2F). Notably, this learning impairment persisted through the subsequent probe trial phase. After 6 days of training, in the tests, Gnb5-CCKO mice took longer

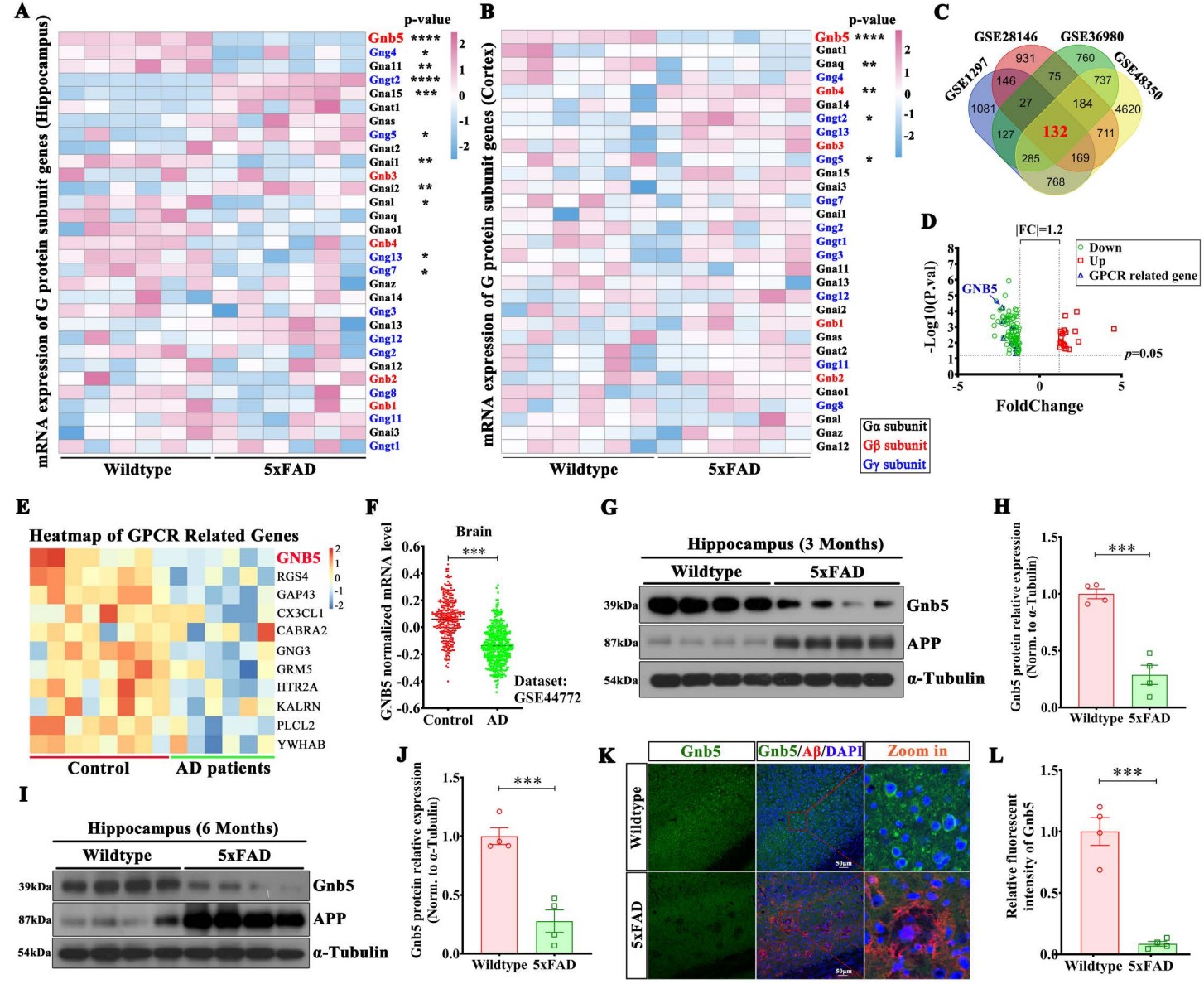

**Fig 1. The Gnb5 expression is reduced in brains of AD patients and 5xFAD mice. (A, B)** The heatmap of G protein family members in the hippocampal **(A)** or cortical **(B)** tissue of 6-month-old Wildtype ($n=6$) and 5xFAD ($n=6$) mice; **(C)** Venn diagram of differentially expressed genes (DEGs) across multiple independent gene expression profile datasets of AD patients; **(D)** Volcano plot of overlapping DEGs in AD patients; red: upregulated; green: downregulated; blue: GPCR-related; **(E)** Heatmap of GPCR-related DEGs obtained from the overlap analysis; **(F)** Expression of Gnb5 in the large sample dataset GSE44772; Control, $n=303$; AD, $n=387$; **(G)** Protein expression levels of Gnb5 and APP in the hippocampal tissue of 3-month-old 5xFAD and Wildtype mice; **(H)** Statistical analysis of Gnb5 protein levels from panel G; Wildtype, $n=4$; 5xFAD, $n=4$; **(I)** Protein expression levels of Gnb5 and APP in the hippocampal tissue of 6-month-old 5xFAD and Wildtype mice; **(J)** Statistical analysis of Gnb5 protein levels from panel I; Wildtype, $n=4$; 5xFAD, $n=4$; **(K)** Immunofluorescence staining of Gnb5 in 6-month-old 5xFAD and Wildtype brain slices; **(L)** Quantitative analysis of Gnb5 fluorescence intensity in 5xFAD and Wildtype groups ($n=4$ mice/group); Individual data points represent mean intensity values per mouse derived from 4 to 6 averaged fluorescence images. Data are presented as mean±SEM; Statistical significance was determined by two-tailed Student $t$ test; ***$P<0.001$.

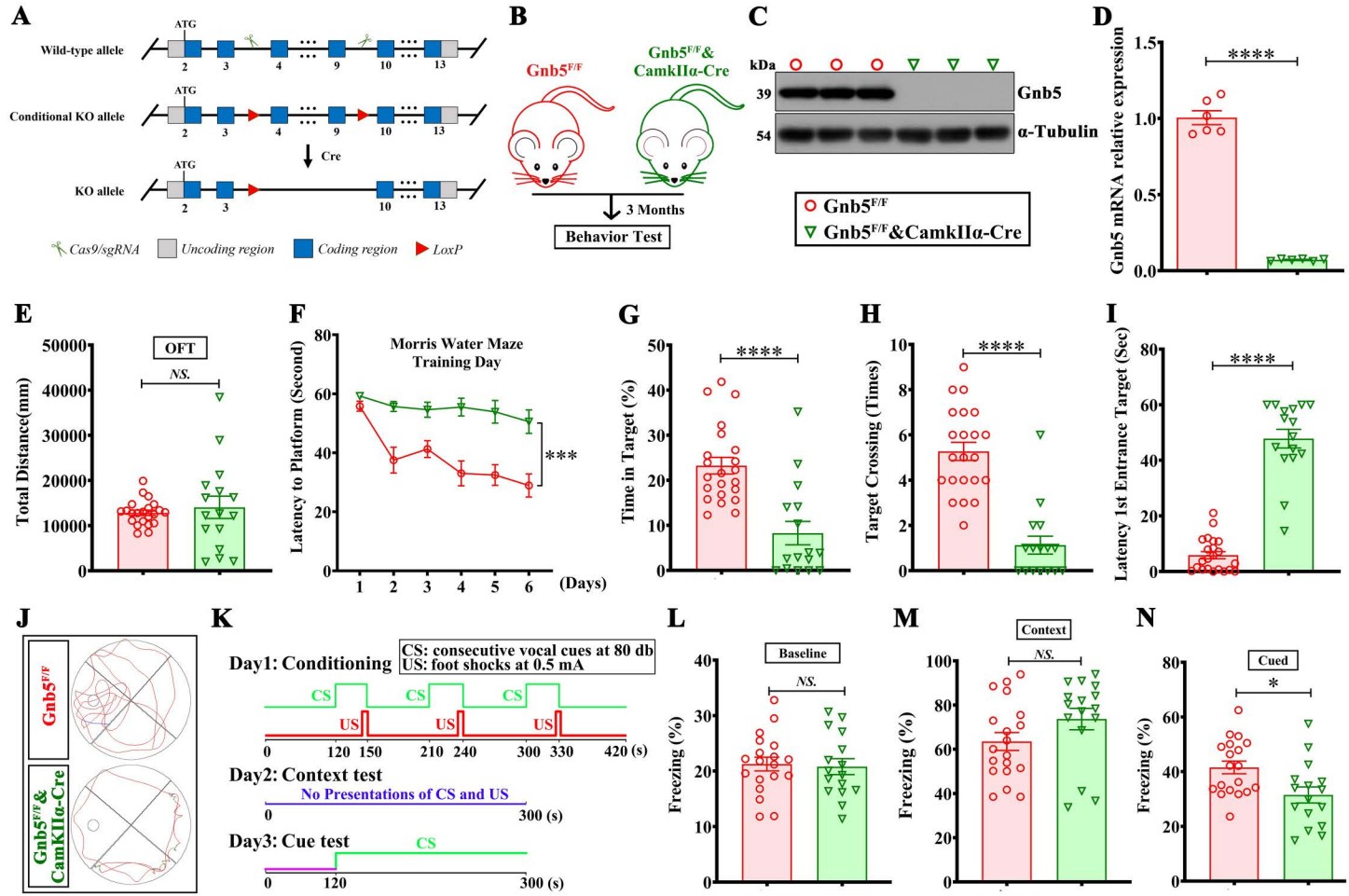

**Fig 2. Excitatory neuron-specific knockout of Gnb5 led to cognitive impairments in mice. (A)** Schematic of Gnb5^F/F mice generation via CRISPR/Cas9; **(B)** Breeding strategy for generating excitatory neuron-specific knockout Gnb5 mice (Gnb5^F/F&CamKIIα-Cre, Gnb5-CCKO); **(C)** western blot validation of Gnb5 knockout in hippocampal tissue; **(D)** qPCR analysis confirming Gnb5 knockout in hippocampal tissue; Gnb5^F/F, $n = 6$; Gnb5-CCKO, $n = 6$; **(E–L)** Animal behavioral tests in mice, Gnb5^F/F, $n = 22$; Gnb5-CCKO, n = 16; (E) Total distance traveled in the open field experiment; (F) Training curve in the Morris water maze (MWM); (G) Percentage of time spent by the mouse in the target quadrant in the MWM; (H) Statistical analysis of the number of times the mouse traverses the target quadrant in the MWM; (I) Latency to first reach the target quadrant in the MWM; (J) Schematic diagram of the mouse's movement trajectory in the water maze; (K–N) Conditioned fear test. (K) Schematic diagram; (L) Baseline freezing percentage of mice in an identical testing context; (M) Percentage of freezing time of mice during the Context phase; (N) Percentage of freezing time of mice during the Cue phase; Data are presented as mean ± SEM; Statistical significance was determined by two-tailed Student $t$ test; NS:($P > 0.05$); *$P < 0.05$; ****$P < 0.0001$.

to locate the platform, exhibited spent less time in the target quadrant, had fewer platform crossings, and longer latency (Fig 2G–2J). In the contextual fear conditioning test (experimental protocol illustrated in Fig 2K), under conditions where no significant intergroup difference in baseline freezing duration was observed in the same environment (Fig 2L), Gnb5-CCKO mice exhibited significantly reduced freezing responses to aversive stimuli compared to Gnb5^F/F controls. This attenuation specifically indicated impaired retention of fear-associated contextual memory. Notably, the memory deficit was more pronounced observed in auditory-cued fear conditioning paradigms (Fig 2N) rather than in contextual fear conditioning assessments (Fig 2M). Together, these results suggest that Gnb5-CCKO mice exhibit cognitive impairment.

## Exogenous expression of Gnb5 in hippocampus ameliorates the cognitive deficits in 5xFAD mice

Given the observed downregulation of *Gnb5* in AD and its negative correlation with Braak stages (S1B Fig), along with evidence that *Gnb5* deficiency in excitatory neurons causes memory deficits (Fig 2), we sought to evaluate whether over-expressing Gnb5 could mitigate cognitive deficits in 5xFAD mice. To achieve this, we constructed an adeno-associated viral (AAV) for Gnb5 expression (AAV2/9-hSyn-Gnb5-P2A-GFP, short for AAV-Gnb5-GFP) and the control virous (AAV2/9-hSyn-P2A-GFP, short for AAV-GFP). These AAVs were injected into the hippocampus of 6-month-old 5xFAD and control mice, respectively (Fig 3A). The efficiency of AAV-Gnb5-GFP expression was confirmed by western blotting and immuno-fluorescence staining (Fig 3B and 3C). Four weeks after injection, we conducted behavioral tests to assess memory and learning function. In the MWM paradigm, mice underwent six consecutive days of training to locate the hidden platform in the target quadrant. Behavioral analysis revealed that while 5xFAD mice receiving AAV-Gnb5-GFP treatment showed significant improvement in learning performance compared to control 5xFAD mice during the training phase (Fig 3D), they demonstrated enhanced spatial memory performance during the probe trial. This improvement was specifically manifested as significantly increased time in the target quadrant and target quadrant crossings compared to 5xFAD controls (Fig 3E–3G). Despite improved cognitive performance, AAV-Gnb5-GFP treatment did not affect locomotor ability, as shown by similar swimming speeds between treated and control mice (Fig 3H). Moreover, in contextual fear conditioning test, baseline freezing levels were comparable across groups, with no significant differences observed in pre-stimulus immo-bility (Fig 3I). Notably, Gnb5-overexpressing 5xFAD mice exhibited higher freezing responses to fear stimuli compared to controls, indicating enhanced memory retention of the aversive context (Fig 3J and 3K). These results indicate that Gnb5 overexpression enhances both spatial and contextual fear memory, functions primarily mediated by the hippocampus. The open field test showed no significant differences in locomotor activity (Fig 3L) or anxiety-like behavior between groups (Fig 3M). Collectively, these findings indicate that Gnb5 overexpression effectively alleviates cognitive deficits in 5xFAD mice.

## Gnb5 inhibits Aβ deposition

Aβ accumulation is a key initiating event in the amyloid cascade hypothesis and contributes to synaptic dysfunction and neurotoxicity in animal models. Given our above results suggesting the role of *Gnb5* in cognitive function, we examined whether it regulates Aβ accumulation. We found that the number or size of Aβ plaques are both decreased in 5xFAD mice injected with AAV-Gnb5-GFP virus compared with control AAV. (Fig 4A–4C).

To further explore the impact of Gnb5 downregulation on Aβ deposition, we generated a triple transgenic model (5xFAD&Gnb5$^{F/F}$&CamKIIα-CreERT) by crossing 5xFAD mice with *Gnb5$^{F/F}$* mice and CamkIIα-CreERT mice, which express tamoxifen-inducible Cre recombinase in excitatory neurons (Fig 4D). The inducible Cre system avoids any developmental deficits that might arise from embryonic *Gnb5* deletion. In 5xFAD mice, amyloid plaques begin to form as early as 2 months of age, and widespread deposition by 3 months [32]. Tamoxifen (100 mg/kg, intraperitoneally) was administrated to 3-month-old 5xFAD mice for 10 days, after two weeks, these mice were subjected to analysis (Fig 4D). We observed a sig-nificant increase in Aβ plaque size and number in the Gnb5 knockdown AD mice compared to control (Fig 4E–4G). These findings indicated that Gnb5 negatively regulates Aβ deposition and targeting Gnb5 could be beneficial for AD therapy.

Additionally, Aβ accumulation triggers the activation of glial cells, leading to neuroinflammation. Thus, overexpression of Gnb5 in AD mice notably attenuated glial activation, reflected in both astrocytes (S3A and S3B Fig) and microglia (S3C and S3D Fig). Intriguingly, Gnb5-CCKO in a non-AD context induces gliosis (S4A–S4D Fig), suggesting a role in modulat-ing neuron-glia crosstalk in the brain.

## Gnb5 regulates BACE1 expression

APP is a transmembrane protein primarily expressed in neurons and undergoes initial cleavage by β-secretase and γ-secretase to generate Aβ peptides [6,33]. To understand how Gnb5 influences the Aβ deposition, we assessed the

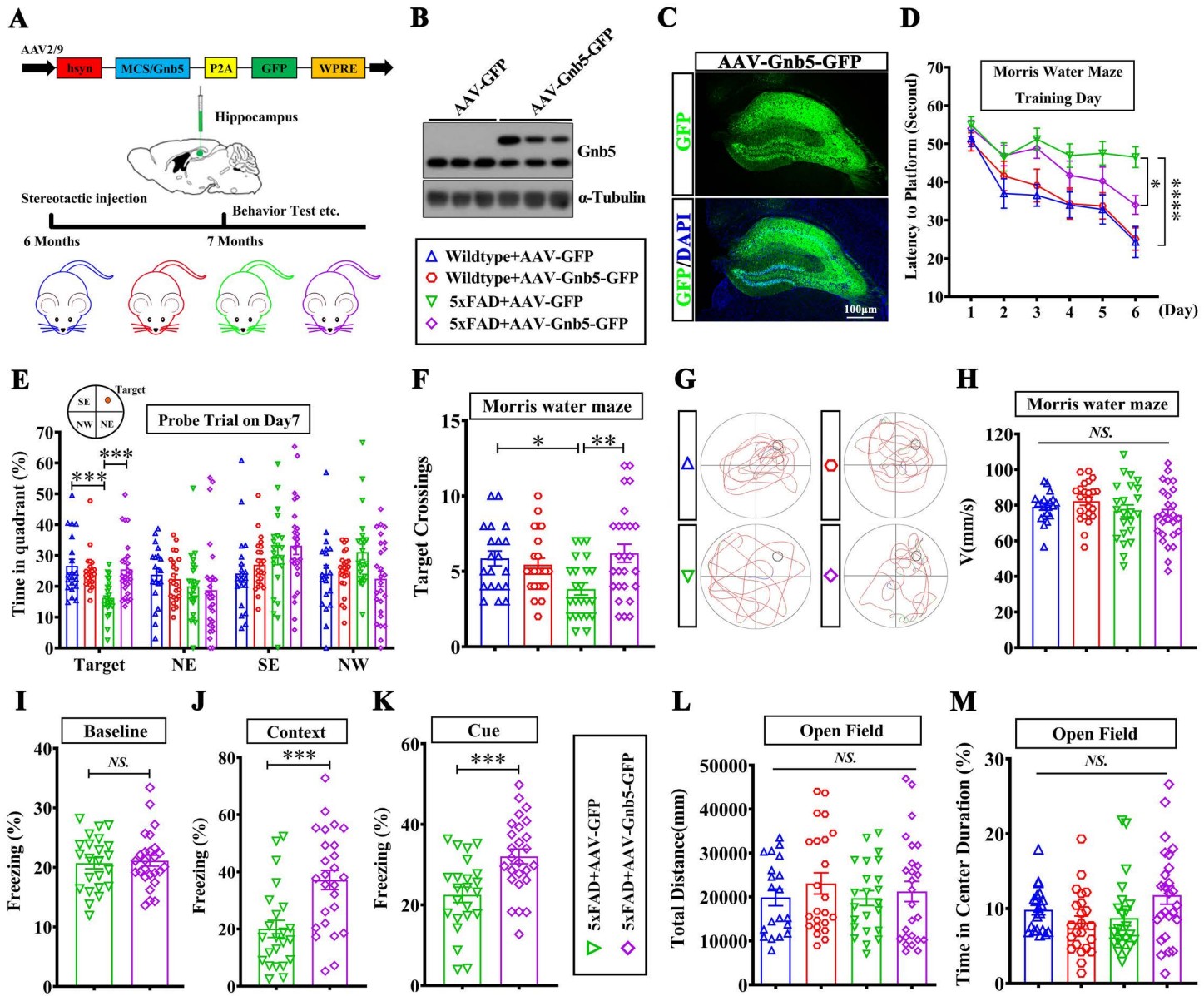

**Fig 3. Exogenous expression of Gnb5 in hippocampus ameliorates the cognitive deficits in 5xFAD mice. (A)** Schematic diagram of AAV-Gnb5 construct and the experimental grouping of AAV-injected mice; **(B)** western blot analysis showing Gnb5 overexpression efficiency after stereotaxic injection of AAV-Gnb5; **(C)** Immunofluorescence images confirming Gnb5 overexpression efficiency in the hippocampus of injected mice; **(D–H)** MWM tests in mice; Wildtype+AAV-GFP, $n = 20$; Wildtype+AAV-Gnb5-GFP, n = 23; 5xFAD+AAV-GFP, $n = 23$; 5xFAD+AAV-Gnb5-GFP, $n = 26$; (D) Training curve in the MWM; (E) Percentage of time spent in the target quadrant in the MWM; (F) Statistical analysis of the number of entries in the target quadrant in the MWM; (G) Schematic diagram of mouse movement trajectory in the MWM; with quadrants marked and platform locations indicated by circles; (H) Movement speed of mice in the MWM; (I-K) The contextual fear conditioning test in mice. **(I)** Baseline freezing percentage of mice in an identical testing context; **(J)** Percentage of freezing time of mice during the Context phase; **(K)** Percentage of freezing time of mice during the Cue phase; Data are presented as mean±SEM; Statistical significance was determined by two-tailed Student $t$ test; NS:($P > 0.05$); ***$P < 0.001$; **(L–M)** The open field test in mice. (L) Total distance traveled in the open field test; (M) Percentage of time spent in the center area during the open field test. Data are presented as mean±SEM; Statistical significance was determined by one-way ANOVA with Holm-Šidák post hoc tests for multiple comparisons; NS:($P > 0.05$); *$P < 0.05$; **$P < 0.01$; ***$P < 0.001$; ****$P < 0.0001$.

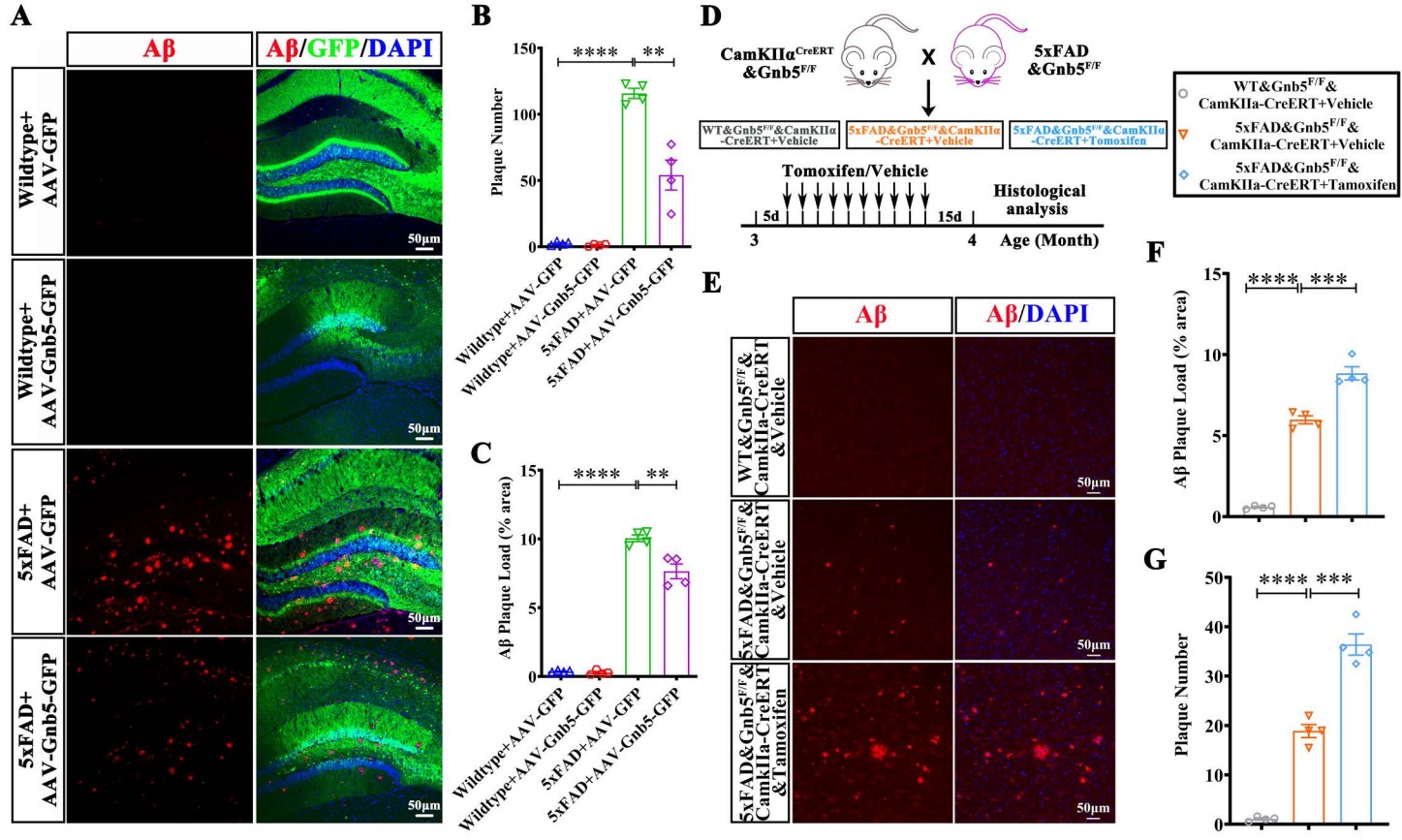

**Fig 4. Gnb5 inhibits the Aβ deposition. (A)** Representative immunofluorescence images of Aβ plaques in the hippocampus sections from Wild-type and 5xFAD mice injected with either AAV-Gnb5 or AAV-GFP; **(B)** Quantification of Aβ plaques number; **(C)** Quantification of Aβ plaques size; **(D)** Schematic diagram of showing the generation of triple transgenic mice with excitatory neuron-specific deletion of Gnb5 in 5xFAD mice, alongside the experimental design; **(E)** Representative immunofluorescence images showing Aβ plaques from indicted mice lines; **(F)** Quantification of Aβ plaque size; **(G)** Quantification of Aβ plaque number; Individual data points represent mean values of per mouse derived from 3 to 5 fluorescence images; Data are presented as mean ± SEM; Statistical significance was determined by one-way ANOVA with Holm-Šidák post hoc tests for multiple comparisons; *$P < 0.01$; ***$P < 0.001$; ****$P < 0.0001$.

expression of key proteins involved in the APP cleavage, including APP, BACE1, Nicastrin, and β-CTF. As expected, 5xFAD mice exhibited elevated levels of APP, BACE1, Nicastrin, and β-CTF. Notably, the protein levels of BACE1 and β-CTF but not Nicastrin were significantly reduced in 5xFAD mice injected with AAV-Gnb5 (Fig 5A and 5B). Conversely, excitatory neuron specific knockdown of Gnb5 in 5xFAD mice did not affect APP and Nicastrin levels but resulted in a marked increase in BACE1 and β-CTF expression (Fig 5C and 5D). Similarly, Gnb5-CCKO mice displayed elevated BACE1 and β-CTF levels compared to *Gnb5^F/F* mice (Fig 5E and 5F).

Having established the regulatory role of Gnb5 in modulating BACE1 protein expression, we next determined whether Gnb5 regulates BACE1 transcription at the mRNA level. Both overexpression and knockdown of Gnb5 had no significant impact on BACE1 mRNA levels, as determined by qPCR (S5A and S5B Fig). Given BACE1 enzyme activity is critical for Aβ generation, we employed a fluorescence resonance energy transfer (FRET)-based assay to quantify BACE1 enzymatic activity. Comparative analysis demonstrated significantly elevated BACE1 activity in 5xFAD mice relative to control animals. Notably, AAV-mediated overexpression of Gnb5 in 5xFAD mice substantially attenuated BACE1 enzymatic activity, restoring it to near-normal levels compared with 5xFAD controls (S5C and S5D Fig). Conversely, BACE1 activity was further increased

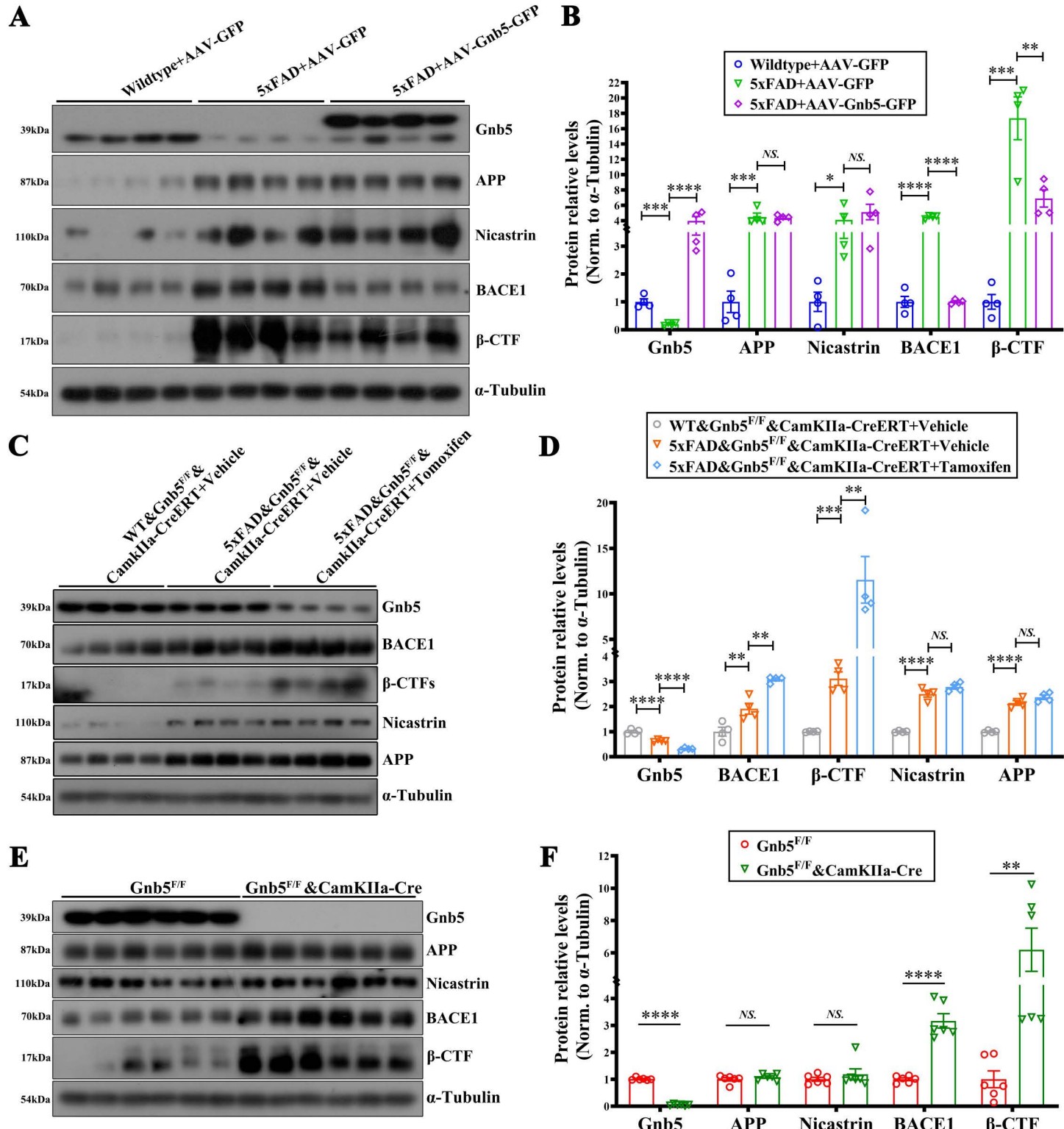

**Fig 5. Gnb5 regulates the BACE1 expression. (A, B)** Overexpression of Gnb5-AAV in 5xFAD mice affects the expression of proteins related to APP cleavage (BACE1, β-CTF, Nicastrin and APP); (A) Representative Western blot data; (B) Statistical analysis of protein levels; *n* = 4, respectively; **(C,**

**D)** Tamoxifen-induced Gnb5 knockdown in 5xFAD mice affects APP-cleaving proteins; (C) Representative Western blot data, (D) Statistical analysis of protein levels; *n* = 4, respectively; **(E, F)** Gnb5-CCKO mice show altered expression of APP-cleaving proteins; (E) Representative Western blot data; (F) Statistical analysis of protein levels; *n* = 6, respectively. Data are presented as mean ± SEM; Statistical significance was determined by one-way ANOVA with Holm–Šidák post hoc tests for multiple comparisons (panel B and D) and two-tailed Student's *t*-test (panel F); *NS*:($P$ > 0.05); ***P* < 0.01; ****P* < 0.001; *****P* < 0.0001.

in 5xFAD mice following Gnb5 knockdown (S5E and S5F Fig). These findings suggest that Gnb5 selectively modulates BACE1 protein level and enzyme activity, likely by acting at the enzymatic cleavage step involved in Aβ generation.

### The animo acid residue Ser81 of Gnb5 is essential for its binding to BACE1

Given the role of Gnb5 in regulating Aβ and BACE1, we next explored whether Gnb5 interacts with BACE1. In vivo co-immunoprecipitation (CO-IP) from brain lysates of *Gnb5^F/F^* and Gnb5-CCKO mice revealed that Gnb5 binds to BACE1 in vivo, as the interaction was diminished in *Gnb5* knockout mice (Fig 6A). To further validate this interaction,

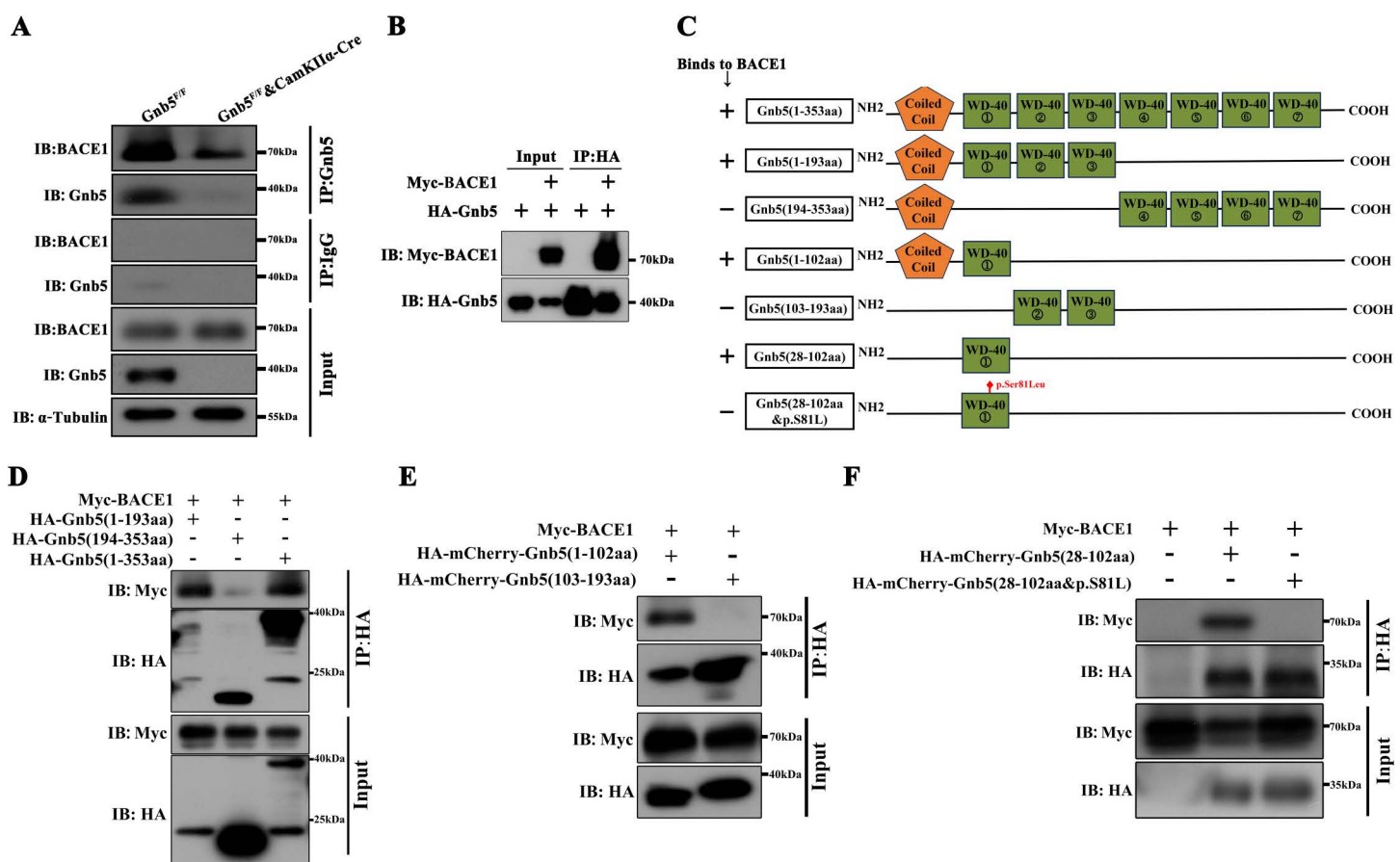

**Fig 6. The animo acid residue Ser81 of Gnb5 is essential for its binds to BACE1. (A)** Co-immunoprecipitation (Co-IP) from brain lysates of Gnb5^F/F and Gnb5-CCKO mice shows endogenous interaction between Gnb5 and BACE1; **(B)** Co-IP assay conducted in HEK293T cells transfected with HA-Gnb5 and Myc-BACE1 confirms exogenous interaction between Gnb5 and BACE1; **(C)** Schematic of the truncation strategy used to generate Gnb5 deletion mutants; with a summary of interactions with BACE1: "+" for positive, "−" for no interaction; **(D)** Co-IP results show the interaction between half-truncated Gnb5 and BACE1; **(E)** Co-IP results show interaction between the quarter-truncated Gnb5 and BACE1; **(F)** Co-IP results show that Gnb5's interaction with BACE1 is mediated by its first WD-40 repeat domain, and the p.S81L mutation disrupts this interaction.

HEK293T cells were co-transfected with Gnb5 and BACE1 expression constructs with tags. The CO-IP further confirmed that Gnb5 interacts with BACE1 in vitro (Fig 6B). Gnb5 comprises an N-terminal coiled-coil domain and seven WD40 domains, which are key for protein-protein interactions within GPCR signaling. To delineate the functional domain of Gnb5 responsible for its interaction with BACE1, we generated several truncated Gnb5 variants (Fig 6C). CO-IP analysis revealed that the N-terminal coiled-coil and first WD40 domain mediate BACE1 binding (Fig 6D and 6E). Furthermore, we confirmed that the first WD domain (28-102aa) is the critical region for their interaction. Notably, the p.Ser81Leu mutation in Gnb5, which has been implicated in cognitive function alterations [18,19], is located within the first WD-repeat domain of Gnb5 (specifically, the 28-102aa region). We then generated the Gnb5 fragment harboring the S81L mutation (28-102aa&p.S81L), which completely disrupted the interaction between Gnb5 with BACE1 (Fig 6F).

### Supplementation of the BACE1-binding fragment of Gnb5 (28-102aa) reduces Aβ deposition in 5xFAD mice

Given the first WD40 domain of Gnb5 (Gnb5 (28-102aa)) is essential for its binding with BACE1 and the S81L mutation disrupts its role, we wondered whether this protein fragment could attenuate AD pathology in vivo. We constructed the AAVs expressed the first WD40 fragment (hSyn-Gnb5 (28-102aa)-GFP), the WD40 fragment with the S81L mutation (hSyn-Gnb5 (28-102aa&p.S81L)-GFP), and also the full-length Gnb5 AAV (hSyn-Gnb5-GFP). These AAVs were injected into the hippocampus of 5-month-old 5xFAD mice for expression 1 month (Fig 7A). We found that the fragmentation Gnb5 (28-102aa) is sufficient to lower the Aβ plaques deposition similar to the full-length AAV-Gnb5 (Fig 7B–7D). In contrast, the AAV-Gnb5 (28-102aa&p.S81L) failed to reduce the Aβ plaques burden (Fig 7B–7D). Consistent with these results, AAV-Gnb5 (28-102aa) reduced the protein levels of BACE1 and β-CTF, whereas the S81L mutant had no effect on these proteins (Fig 7E and 7F). To further investigated whether the interaction between Gnb5 and BACE1 affects downstream signaling, we examined two established Gnb5 effectors: Rgs7 [34–36] and Akt [37–39]. Western blot analysis showed no significant changes in Rgs7, Akt, or p-Akt levels following overexpression of either Gnb5 (28–102aa)-AAV or its S81L mutant (S6A and S6B Fig). Similarly, BACE1 knockdown using siR1-BACE1 (75% efficiency; S6C Fig) did not alter these protein levels in HEK293T cells (S6D and S6E Fig). These findings suggest that Gnb5 regulates BACE1 through Ser81 without engaging canonical downstream pathways involving Rgs7 or Akt. Together, these results suggest that Gnb5 regulates Aβ production by interacting with BACE1 through the first WD-40 repeat domain, with the Ser81 residue being critical for this regulation.

## Discussion

G-protein-coupled receptors (GPCRs) play a crucial role in Alzheimer's disease (AD) pathophysiology, including promoting Aβ and tau accumulation, driving neuroinflammation, and impairing synaptic dysfunction [40–42]. Given their central regulatory positions in neuronal signaling, GPCRs and their downstream effectors have emerged as promising therapeutic targets for AD intervention. For example, serotonin receptors (5-HT4R) and muscarinic acetylcholine receptors mediate Aβ deposition and tau pathology, positioning them as promising therapeutic targets [43,44]. Clinically approved acetylcholinesterase inhibitors, such as Donepezil, indirectly boost GPCR signaling by increasing acetylcholine levels. In addition, newer approaches aim to directly target GPCRs, such as cannabinoid receptors (CB1, CB2) for neuroprotection and metabotropic glutamate receptors for synaptic enhancement are actively under investigation [45–47]. In this context, our study identifies Gnb5, a brain-enriched Gβ subunit of the GPCR complex, as a novel negative regulator of BACE1-mediated Aβ generation and a critical modulator of AD progression.

Gnb5 is predominantly expressed in neurons and plays an important role in various intracellular signaling processes, with emerging evidence linking it to the pathogenesis of multiple neurological disorders [14,18,20]. Recent genetic studies have implicated *Gnb5* as a candidate risk gene for AD, with experimental evidence showing that heterozygous Gnb5 enhances amyloid plaque formation and neurofibrillary tangle development in AD model mice [20]. Furthermore,

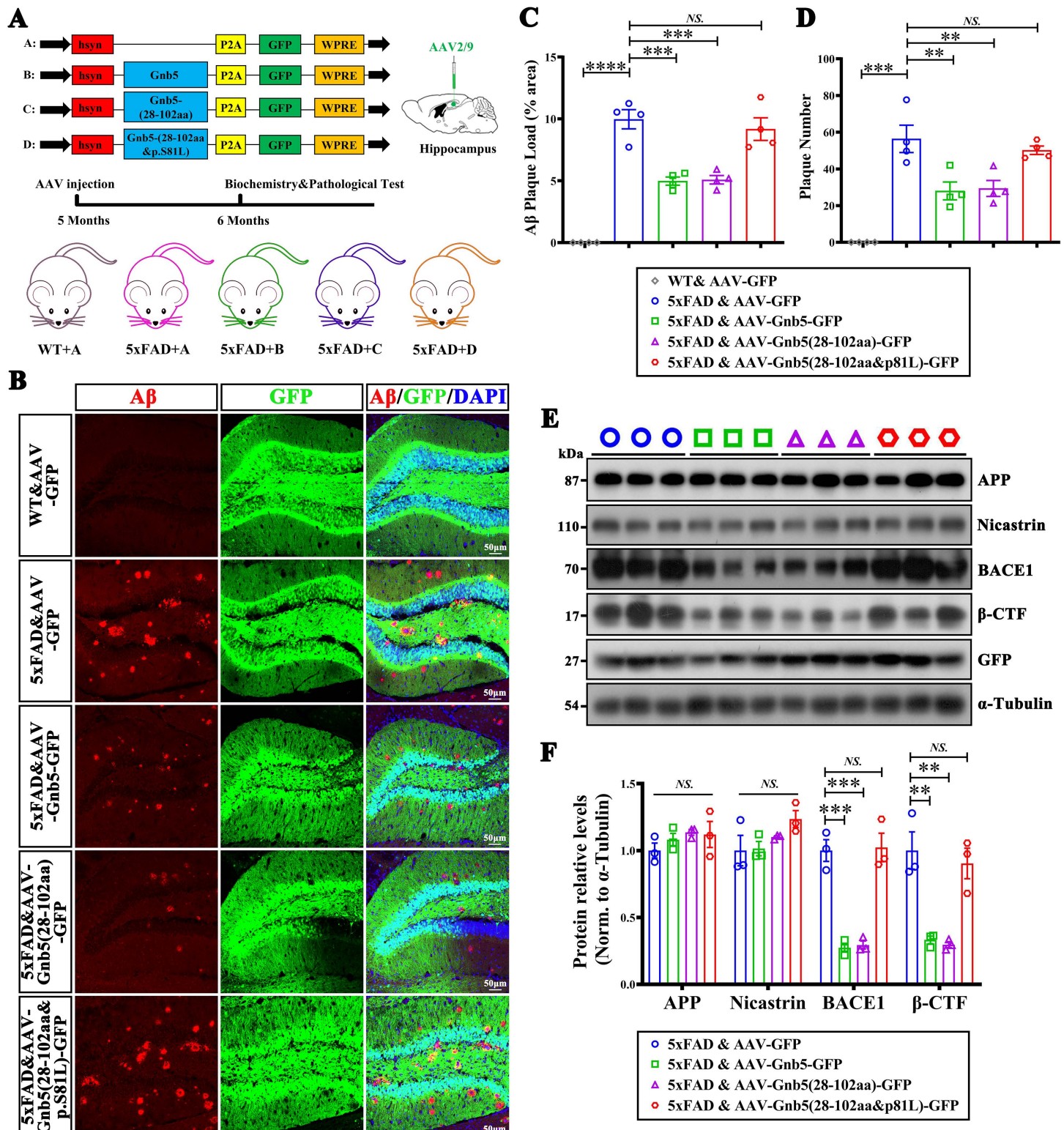

**Fig 7. The protein fragment Gnb5 (28-102aa) alleviated the AD pathology, while the p.S81L mutant abolished this effect. (A)** Schematic of the overexpression AAV constructs and experimental grouping; **(B)** Aβ deposition staining of the overexpressed full-length Gnb5, truncated fragments, and

mutant truncated fragment AAV in 5xFAD mice; **(C, D)** Quantification of Aβ plaques size **(C)** or Aβ plaques number **(D)**; Individual data points represent mean values of per mouse derived from 3-5 fluorescence images; **(E, F)** Effect of overexpressing full-length Gnb5, truncated fragments, and mutant truncated fragment viruses on the expression of proteins related to APP cleavage process in 5xFAD. Representative western blot images **(E)** and statistical analysis of protein levels **(F)**; $n = 3$, respectively. Data are presented as mean ± SEM; Statistical significance was determined by one-way ANOVA with Holm-Šidák post hoc tests for multiple comparisons; NS:($P > 0.05$); **$P < 0.01$; ***$P < 0.001$; ****$P < 0.0001$.

homozygous knockout of *Gnb5* leads to significant neurodevelopmental impairments [16]. Despite these associations, the precise role of Gnb5 in AD pathophysiology remains poorly understood.

Our findings reveal that Gnb5 is significantly downregulated in the brains of both AD patients and 5xFAD mouse models. While the decrease in Gnb5 expression could be a downstream effect of disease progression, our results suggest the intriguing possibility of a bidirectional regulatory relationship between Gnb5 and Aβ pathology. Specifically, Aβ accumulation might contribute to the suppression of Gnb5 expression, potentially establishing a self-perpetuating pathological loop. Supporting this notion, neuron-specific deletion of Gnb5 exacerbates Aβ plaque accumulation and leads to severe cognitive impairments, while hippocampal overexpression of Gnb5 reverses these effects. These observations support a potential contributory role of Gnb5 loss in AD pathogenesis, although the exact causal relationships remain to be fully elucidated.

BACE1 is crucial for APP cleavage, leading to Aβ production, and its activity is positively correlated with AD pathology [48]. Thus, inhibiting BACE1 has been considered a promising therapeutic strategy to reduce Aβ production, alleviate neurotoxicity, and improve cognitive function. Numerous preclinical studies and clinical trials have shown that BACE1 inhibitors can significantly reduce brain Aβ levels and improve cognitive function [49,50]. Importantly, early intervention with BACE1 inhibitors could potentially delay AD progression by reducing early Aβ accumulation [7,51,52]. However, the clinical application of these inhibitors faces numerous limitations and challenges, particularly regarding safety, as global BACE1 inhibition cause side effects, including but not limited to gastrointestinal discomfort, liver function abnormalities, and neurological adverse reactions [9,10]. Therefore, identifying novel regulatory factors of BACE1 in a more selective and controlled manner could enhance the efficacy and safety of AD treatments that target BACE1 activity. Given its predominant neuronal expression, Gnb5 represents an attractive candidate for selectively modulating BACE1 activity within the brain, potentially minimizing off-target effects in peripheral tissues.

Beyond its canonical role in GPCR signaling, Gnb5 exerts broader functional effects. The S81L point mutation in Gnb5 has been previously linked to a spectrum of neurodevelopmental disorders, including language developmental disorders, ADHD, and motor delays and epilepsy [14,18,19]. Structural studies suggest that the S81L mutation likely changes the conformation of Gnb5, as the hydroxyl side chain of Ser81 plays a critical role in molecular interactions, whereas the hydrocarbon side chain of Leu introduces conformational changes at the interaction site [53]. Such alterations may disrupt Gnb5 localization and interactions with other signaling molecules, potentially impacting Gnb5-mediated signaling and warranting further investigation. In line with this, we demonstrate that the S81L mutation abolishes Gnb5-BACE1 interaction (Fig 6), reinforcing its critical role in regulating BACE1 expression and potentially contributing to AD pathophysiology (Fig 7). Interestingly, despite Gnb5's structural classification as a Gβ subunit, its overexpression—or overexpression of its BACE1-interacting fragment—did not significantly affect the levels of classical downstream effectors such as Rgs7 or Akt. This suggests that Gnb5 may regulate BACE1 expression through non-canonical mechanisms independent of conventional GPCR signaling pathways, an area that warrants deeper investigation.

While our study provided significant insights into the Gnb5 role in the AD, several limitations should be acknowledged. A key question that remains unanswered is how Gnb5, a G protein, can regulate the expression of BACE1. Although our study did not fully address this mechanism, several pathways warrant further investigation, including G protein-dependent signaling pathways, BACE1 degradation, and intracellular trafficking. Additionally, while Gnb5 itself may not be an ideal druggable target as a G protein subunit, its effectiveness in reducing Aβ load and improving cognitive performance in AD mouse

models highlights the potential of targeting regulators upstream or downstream in Gnb5-signaling pathways. This underscores the importance of delineating the specific molecular pathways associated with Gnb5 in the brain. Moreover, while our data demonstrate that Gnb5 interacts with BACE1 and is associated with reduced Aβ production, the cognitive improvements observed in 5xFAD mice are unlikely to be solely attributed to BACE1 modulation. G protein activation, including that mediated by Gnb5, modulates multiple intracellular signaling pathways that involve second messengers such as cAMP and calcium, as well as downstream kinases like PKA and PKC [54,55]. These pathways may independently contribute to the cognitive improvements observed [56–59], suggesting that Gnb5 may influence AD-related phenotypes through both BACE1-dependent and BACE1-independent mechanisms. Finally, in the use of the 5xFAD mouse model, which—while valuable for studying Aβ-driven pathology—does not fully recapitulate the complexity of human AD, particularly the progressive tau pathology and sporadic disease onset seen in most patients [32]. To broaden the translational relevance of our findings, future studies should incorporate additional models, including APP knock-in lines with physiological expression, dual-transgenic models incorporating tau pathology, and validation in human postmortem brain tissues.

In summary, our study demonstrates the significant impact of Gnb5 on cognitive function and Aβ plaque deposition, elucidating the mechanisms by which it regulates BACE1, which is central to Alzheimer's disease progression.

## Materials and methods

### Animals

All mice were housed in the Laboratory Animal Center of Xiamen University, and all experimental procedures involved were performed according to protocols approved by the Institutional Animal Care and Use Committee of Xiamen University (Approval Number: XMULAC20200054) and were conducted in compliance with the Biosafety Law of the People's Republic of China, the Regulations on the Administration of Experimental Animals, the National Standards for Experimental Animals (GB14925-2010), the Guidelines for Ethical Review of the Welfare of Experimental Animals (GB/T 35892-2018), and institutional guidelines. Mice were maintained under a 12-hour light/dark cycle with ad libitum access to standard rodent chow and water. Wild-type C57BL/6J mice were directly obtained from the Xiamen University Laboratory Animal Center. The 5xFAD mice [32] were provided by Model Animal Research Center of Nanjing. The Floxed Gnb5 mouse strain (Gnb5$^{F/F}$) were generated by GemPharmatech Co. CamKIIα-Cre mice [60] and CamKIIα-CreERT mice [61] were obtained from Jackson laboratory (Bar Harbor, ME).

### GEO dataset information

The GSE1297 dataset compared hippocampal gene expression between nine control subjects and seven individuals with severe AD. The GSE28146 dataset used laser capture microdissection to exclude major white matter tracts, CA1 hippocampal gray matter was selectively isolated from formalin-fixed, paraffin-embedded sections, comprising eight controls and seven severe AD cases. The GSE36980 dataset analyzed hippocampal gene expression from post-mortem AD brains, including 10 controls and 7 AD subjects. The GSE48350 dataset contain microarray data from the human hippocampus were acquired, containing 32 control samples and 31 AD cases. The GSE44772 contain multi-tissue gene expression profiles of human brain, including 303 controls and 387 AD subjects. Differential expression analysis of cross-platform normalized dataset, containing multiple brain regions, downloaded and analyzed from the AD database (http://www.alzdata.org/index.html) [62,63].

### Generation of adeno-associated virus (AAVs) and stereotactic injection of AAVs

All viral constructs included pAAV-hsyn-Gnb5-3xFLAG-GFP (titer: 6.89 × 10$^{12}$ vg/mL), pAAV-hSyn-Gnb5 (28-102aa)-3xFLAG-GFP (titer: 1.19 × 10$^{13}$ vg/mL), pAAV-hSyn-Gnb5 (28-102aa-S81L)-3xFLAG-GFP (titer: 6.31 × 10$^{12}$ vg/mL), and pAAV-hSyn-3xFLAG-GFP (titer: 1.13 × 10$^{13}$ vg/mL). Mice were anesthetized with pentobarbital (20 mg/kg, i.p.) prior to stereotaxic injection of viruses. Stereotaxic injections were performed targeting the hippocampal region using the

following coordinates relative to Bregma: anterior-posterior (AP)—2.0 mm, medial-lateral (ML) ±1.5 mm, and dorsal-ventral (DV)—1.6 mm from the brain surface. Behavioral assessments commenced four weeks post-injection. Upon completion of behavioral testing, animals were euthanized followed by brain collection for subsequent histological analysis, including immunofluorescence staining and western blot experiment.

## Co-immunoprecipitation

Transfected HEK293T cells were lysed in ice-cold buffer containing 20 mM Tris-HCl (pH 7.4), 100 mM NaCl, 1 mM EDTA, 0.5% NP-40, and Complete Protease Inhibitor Cocktail (Roche, #04693132001). For immunoprecipitation, cell lysates were incubated with either anti-Myc (Thermo Fisher Scientific, #132500; 1:200 dilution) or anti-HA (Sigma-Aldrich, #H6908; 1:200 dilution) antibodies for 12 h at 4 °C, followed by 2 h r incubation with Dynabeads Protein G (Thermo Fisher Scientific, #10004D). The immunoprecipitates were then analyzed by western blotting using standard protocols.

## BACE1 activity assay

The enzymatic activity of BACE1 was quantified using a FRET-based assay kit (Cat# P0362S; Beyotime Biotechnology). The substrate consists of a BACE1-specific peptide flanked by a fluorogenic donor (MCA) and a quencher (Dnp). In the intact substrate, FRET occurs between MCA and Dnp due to their proximity (7–10 nm), resulting in suppressed fluorescence. Upon BACE1-mediated cleavage, the spatial separation of MCA and Dnp abolishes FRET, allowing MCA fluorescence emission. The assay was performed according to the manufacturer's protocol. Briefly, Hippocampal tissues were homogenized in ice-cold lysis buffer and centrifuged at 12,000$g$ for 15 min at 4 °C. Supernatants were incubated with BACE1 substrate in black 96-well plates. Fluorescence measurements were performed using a Varioskan Flash full-wavelength multimode microplate reader (Thermo Fisher Scientific) with excitation/emission wavelengths set to 325/393 nm. Measurements were taken at 10-minute intervals over 60 min. Relative fluorescence units were normalized to protein concentration and expressed as BACE1 enzymatic activity (nmol/min/mg).

## Animal behavioral tests

All mice used in behavioral testing were 6 months old and group-housed with five animals per cage. To ensure environmental acclimation, mice were transferred to the testing rooms 2 h prior to experimentation. Behavioral assessments were conducted on alternating days using validated equipment. Cognitive functions, including learning and memory were evaluated through a standardized battery of tests: the MWM and fear conditioning chamber.

## Open field test

Locomotor activity and spontaneous exploration patterns were quantified using an open-field test paradigm. Behavioral assessments were conducted in a polycarbonate arena (40 × 40 × 30 cm) monitored by SMART 3.0 video tracking software. Each mouse underwent a 10-min acclimatization period followed by a 10-min recording session. Movement parameters, including total ambulatory distance and center zone occupancy (20 × 20 cm inner quadrant) were automatically calculated by the tracking system.

## Morris water maze (MWM)

The MWM apparatus consisted of a blue circular tank (120 cm diameter) maintained at 22 ± 0.5 °C, containing a submerged platform (1 cm below water surface) in the target quadrant. Distinct geometric patterns served as visual cues on the four cardinal walls. Animal movements were tracked using an overhead camera connected to CleverSys TopScanLite software (Clever Sys). During the 6-day training phase, mice underwent twice-daily trials with randomized entry quadrants. Each trial permitted 60-s platform searches, followed by 20-s platform habitation and 120-s inter-trial recovery.

 

Animals failing to locate the platform within 60 s were gently guided to it. For probe testing on day 7, the platform was removed. Mice were released from the NE quadrant (opposite quadrant) and allowed 60-s free navigation. Spatial memory was quantified by platform zone crossings, quadrant occupancy time (%) and first latency to target area.

### Fear conditioning test

This study uses a fear conditioning system to investigate environmental conditioned fear in mice. First, a non-transparent chamber with a removable grid floor made of stainless-steel bars is prepared. Software is used to control auditory tones and mild foot shocks. On the first training day, mice were placed in the chamber and allowed 120 s of acclimatization before receiving foot shocks (0.5 mA, 2 s) followed by three consecutive auditory cues (80 dB, 30 s each). This conditioning paradigm was repeated twice with 10-min intervals. During testing on day 2, mice were re-exposed to the chamber for 5 min without stimulation. On the final test day (day 3), mice were re-exposed to the conditioned fear chamber with modified contextual cues. Following a 120-s habituation period, animals were subjected to an 80 dB acoustic stimulus lasting 180 s. Freezing behavior was continuously monitored throughout the auditory stimulation period, with both absolute freezing duration and its percentage relative to the total stimulus exposure time being systematically recorded.

### Experimental design

All experiments in this study were conducted with at least three mice per experimental group or repeated in three independent biological replicates. All mouse-related procedures and data analyses were performed using blinded protocols. Specifically, researchers were kept unaware of mouse genotypes throughout behavioral testing and subsequent analyses.

### Statistical analysis

Data are presented as means ± SEM. Statistical analyses were performed using GraphPad Prism 8.0 (GraphPad Software, Prism software for PC) with unpaired two-tailed Student $t$ test for comparisons between two groups, or one-way ANOVA with Holm-Šidák post hoc tests for multiple comparisons. Statistical significance thresholds were defined as: NS. (not significant): $P > 0.05$, $*P < 0.05$, $**P < 0.01$, $***P < 0.001$, and $****P < 0.0001$.

### Consent statement

All animal studies were performed according to the protocols approved by the Institutional Animal Care and Use Committee of Xiamen University. The data sets used and/or analyzed during the current study are available from the corresponding author.

### Supporting information

**S1 Text. Inventory of Supplementary Information: Detailed Supplemental Methods and Materials and Table of Primer sequences used in this study.**
(DOCX)

**S1 Fig. The reduction of Gnb5 expression is closely related to the occurrence and development of AD, related to Fig 1. (A)** Pearson correlation analysis of Gnb5 expression with the age of AD patients in a large sample dataset (GSE44772), $n = 690$; **(B)** Pearson correlation analysis of Gnb5 expression with the Braak stage classification of AD patients, $n = 704$; **(C)** Pearson correlation analysis of Gnb5 expression with the Frontal atrophy classification of AD patients, $n = 721$; **(D–F)** Cross-platform integration of multiple datasets showing Gnb5 expression in different brain regions of AD patients; (D) Entorhinal Cortex, Control, $n = 39$; AD, $n = 39$; (E) Frontal Cortex, Control, $n = 128$; AD, $n = 104$; (F) Temporal Cortex, Control, $n = 39$; AD, n = 52; **(H)** Statistical analysis of Gnb5 protein levels in panel **G**, $n = 4$, respectively; **(J)**

Statistical analysis of Gnb5 protein levels in panel *I*, *n* = 4, respectively; **(L)** Statistical analysis of Gnb5 expression in panel **K**; *n* = 4, respectively.
(TIF)

**S2 Fig. The Gnb5 expression localization and genotyping of excitatory neuron-specific Gnb5 knockout mice, related to** Fig 2. **(C)** Statistical analysis of Gnb5 protein levels from panel B; n = 4, respectively; **(D)** mRNA levels of Gnb5 in neurons (*n* = 12), astrocytes (*n* = 6), and microglia (*n* = 6); **(J)** Comparison of body weight between Gnb5-CCKO (*n* = 20) and Gnb5F/F (*n* = 15) mice; **(K)** Time spent on the rotarod, Gnb5-CCKO (*n* = 21) and Gnb5F/F (*n* = 16) mice.
(TIF)

**S3 Fig. The Gnb5 overexpression rescued the abnormal expression of synaptic proteins and gliosis in 5xFAD mice, related to** Fig 4. **(B, D)** Statistical analysis of the number of astrocytes (B) or microglia (D) in the hippocampus of 6-month-old AAV-GFP or AAV-Gnb5-GFP injected Wild-type and 5xFAD mice; *n* = 3, respectively.
(TIF)

**S4 Fig. Effects of Gnb5-CCKO mice on gliosis, related to** Fig 4. **(B, D)** Statistical analysis of the number of astrocytes (B) or microglia (D) in the hippocampus of 6-month-old Gnb5F/F and Gnb5-CCKO mice; *n* = 3, respectively.
(TIF)

**S5 Fig. Effects of Gnb5 on the mRNA expression and activity level of BACE1, related to** Fig 5. **(A)** Overexpression of Gnb5-AAV in the hippocampus to detect BACE1 mRNA expression; *n* = 6, respectively; **(B)** The mRNA expression level of BACE1 in the hippocampus of Gnb5-CCKO mice; *n* = 6, respectively; **(C)** Fluorescence intensity of BACE1-mediated fluorogenic products was measured in hippocampal tissues from 5xFAD mice receiving AAV-mediated Gnb5 overexpression; Kinetic readings were recorded at 10-min intervals over a 60-min assay period; Information including the average (Avg), SEM, and sample size; **(D)** Quantitative analysis of BACE1 enzymatic activity (nmol/min/mg) from panel C; *n* = 6 mice/group; **(E)** Tamoxifen-inducible Gnb5 knockdown was performed in hippocampal tissues of 5xFAD mice, with fluorescence intensity of BACE1-generated products quantified every 10 min for 60 min; Information including the Avg, SEM, and sample size; **(F)** Quantitative analysis of BACE1 enzymatic activity (nmol/min/mg) from panel E; *n* = 4 mice/group.
(TIF)

**S6 Fig. The effect of the Ser81 mutation of Gnb5 and the knockdown of BACE1 on the activation of Gnb5 downstream signaling, related to** Fig 7. **(B)** The effects of viral-delivered Gnb5 truncated fragments or mutant truncated fragments on Rgs7, Akt and p-Akt expression in 5xFAD mice; Quantitative analysis of the western blot data shown in panel **A**; *n* = 4, respectively; **(C)** qPCR analysis was performed to evaluate the knockdown efficiencies of three BACE1-targeting siRNAs in HEK293T cells; *n* = 4, respectively; **(E)** HEK293T cells transfected with siR1-BACE1 to assess the protein levels of BACE1, Rgs7, Akt and p-Akt; Quantitative analysis of the western blot data shown in panel **D**; *n* = 4, respectively.
(TIF)

**S1 Raw Images. Unedited blot and gel images.**
(PDF)

**S1 Table. Excel spreadsheet containing detailed data matrices supporting all figures in the study.** Complete annotations for the tabular data are presented below. **Tab Fig 1: (A)** The heatmap data of G protein family members in the hippocampal tissue of 6-month-old Wildtype (*n* = 6) and 5xFAD (*n* = 6) mice; **(B)** The heatmap data of G protein family members in the cortical tissue of 6-month-old Wildtype (*n* = 6) and 5xFAD (*n* = 6) mice; **(C)** The data in the overlapping part of the Venn diagram (132 elements); **(D)** The data information for creating volcano plot; **(E)** The data information for creating heatmap of GPCR-related DEGs; **(F)** Expression of Gnb5 in the large sample dataset GSE44772; Control, *n* = 303; AD, *n* = 387; **(H)** Statistical analysis of Gnb5 protein levels from panel **G**; Wildtype, *n* = 4; 5xFAD, *n* = 4; **(J)** Statistical analysis

of Gnb5 protein levels from panel I; Wildtype, $n = 4$; 5xFAD, $n = 4$; **(L)** Quantitative analysis of Gnb5 fluorescence intensity in 5xFAD and Wildtype groups; Wildtype, $n = 4$; 5xFAD, $n = 4$. **Tab Fig 2: (D)** qPCR data of Gnb5 knockout in hippocampal tissue; Gnb5$^{F/F}$, $n = 6$; Gnb5-CCKO, $n = 6$; **(E–I, L–N)** Animal behavioral tests in mice, Gnb5$^{F/F}$, $n = 22$; Gnb5-CCKO, $n = 16$; **(E)** Total distance traveled in the open field experiment; **(F)** Training curve in the Morris water maze (MWM); **(F-day6)** Data from the sixth day of MWM training; **(G)** Percentage of time spent by the mouse in the target quadrant in the MWM; **(H)** Statistical analysis of the number of times the mouse traverses the target quadrant in the MWM; **(I)** Latency to first reach the target quadrant in the MWM; **(L)** Baseline freezing percentage of mice in an identical testing context; **(M)** Percentage of freezing time of mice during the Context phase; **(N)** Percentage of freezing time of mice during the Cue phase. **Tab Fig 3: (D–F, H)** MWM tests in mice; Wildtype+AAV-GFP, $n = 20$; Wildtype+AAV-Gnb5-GFP, $n = 23$; 5xFAD+AAV-GFP, $n = 23$; 5xFAD+AAV-Gnb5-GFP, $n = 26$; **(D)** Training curve in the MWM; **(D-day6)** Data from the sixth day of MWM training; **(E)** Percentage of time spent in the target quadrant in the MWM; **(F)** Statistical analysis of the number of entries in the target quadrant in the MWM; **(H)** Movement speed of mice in the MWM; **(I–K)** The contextual fear conditioning test in mice; 5xFAD+AAV-GFP, $n = 23$; 5xFAD+AAV-Gnb5-GFP, $n = 26$; **(I)** Baseline freezing percentage of mice in an identical testing context; **(J)** Percentage of freezing time of mice during the Context phase; **(K)** Percentage of freezing time of mice during the Cue phase; **(L)** Total distance traveled in the open field test; **(M)** Percentage of time spent in the center area during the open field test. **Tab Fig 4: (B, C)** Quantification of Aβ plaques in the hippocampus sections from Wildtype and 5xFAD mice injected with either AAV-Gnb5 or AAV-GFP; Wildtype+AAV-GFP, $n = 4$; Wildtype+AAV-Gnb5-GFP, $n = 4$; 5xFAD+AAV-GFP, $n = 4$; 5xFAD+AAV-Gnb5-GFP, $n = 4$; **(B)** Quantification of Aβ plaques number; **(C)** Quantification of Aβ plaques size; **(F, G)** Quantification of Aβ pylaques from indicted mice lines; WT&Gnb5$^{F/F}$&CamKIIa-CreERT+Vehicle, $n = 4$; 5xFAD&Gnb5$^{F/F}$&CamKIIa-CreERT+Vehicle, $n = 4$; 5xFAD&Gnb5$^{F/F}$&CamKIIa-CreERT+Tamoxifen, $n = 4$; **(F)** Quantification of Aβ plaque size; **(G)** Quantification of Aβ plaque number. **Tab Fig 5: (B)** Overexpression of Gnb5-AAV in 5xFAD mice affects the expression of proteins related to APP cleavage (BACE1, β-CTF, Nicastrin and APP); Statistical analysis of protein levels; $n = 4$, respectively; **(D)** Tamoxifen-induced Gnb5 knockdown in 5xFAD mice affects APP-cleaving proteins; Statistical analysis of protein levels; $n = 4$, respectively; **(F)** Gnb5-CCKO mice show altered expression of APP-cleaving proteins; Statistical analysis of protein levels; $n = 6$, respectively. **Tab Fig 7: (C, D)** Quantification of Aβ plaques in the overexpressed full-length Gnb5, truncated fragments, and mutant truncated fragment AAV in 5xFAD mice; $n = 4$, respectively; **(C)** Quantification of Aβ plaques size; **(D)** Quantification of Aβ plaques number; **(F)** Effect of overexpressing full-length Gnb5, truncated fragments, and mutant truncated fragment viruses on the expression of proteins related to APP cleavage process in 5xFAD; Statistical analysis of protein levels; $n = 3$, respectively.
(XLSX)

## Author contributions

**Conceptualization:** Jie Zhang.

**Data curation:** Shaokun Chen, Shuzhong Wang, Erqu Chen, Huifang Li.

**Formal analysis:** Shaokun Chen, Shuzhong Wang, Erqu Chen.

**Funding acquisition:** Jie Zhang.

**Investigation:** Shaokun Chen, Kai Zhuang, Chensi Liang, Dan Can.

**Methodology:** Shaokun Chen, Jiechao Zhou, Erqu Chen, Kai Zhuang, Chensi Liang, Jing Li.

**Project administration:** Shaokun Chen, Jiechao Zhou.

**Resources:** Shaokun Chen, Jie Zhang.

**Software:** Shaokun Chen, Shuzhong Wang, Chensi Liang, Dan Can, Huifang Li.

**Supervision:** Jiechao Zhou, Jing Li, Jie Zhang.

**Validation:** Shaokun Chen, Erqu Chen, Kai Zhuang, Raozhou Lin.

**Visualization:** Shaokun Chen, Jiechao Zhou, Raozhou Lin.

**Writing – original draft:** Shaokun Chen, Jiechao Zhou, Jie Zhang.

**Writing – review & editing:** Shaokun Chen, Jiechao Zhou, Raozhou Lin, Jie Zhang.

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
