## [Editor Report · Decision Letter 0]

Dear Dr Zhang,

Thank you for submitting your manuscript entitled "Gnb5 ameliorates cognitive deficits by reducing β-amyloid deposition through BACE1 in Alzheimer's disease" for consideration as a Research Article by PLOS Biology.

Your manuscript has now been evaluated by the PLOS Biology editorial staff and I am writing to let you know that we would like to send your submission out for external peer review.

Once your full submission is complete, your paper will undergo a series of checks in preparation for peer review. After your manuscript has passed the checks it will be sent out for review. To provide the metadata for your submission, please Login to Editorial Manager (https://www.editorialmanager.com/pbiology) within two working days, i.e. by Dec 20 2024 11:59PM.

**As a heads up, please note that the PLOS Biology editorial office will be closed from Dec 23 - Jan 3rd over the Holidays. I will therefore likely not be able to start the reviewer search until early in January. I apologize in advance for the delay!

As an additional note, if your manuscript has been previously peer-reviewed at another journal, PLOS Biology is willing to work with those reviews in order to avoid re-starting the process. Submission of the previous reviews is entirely optional and our ability to use them effectively will depend on the willingness of the previous journal to confirm the content of the reports and share the reviewer identities. Please note that we reserve the right to invite additional reviewers if we consider that additional/independent reviewers are needed, although we aim to avoid this as far as possible. In our experience, working with previous reviews does save time.

Kind regards,

Luke

Lucas Smith, Ph.D.

Senior Editor

PLOS Biology

lsmith@plos.org

---

## [Decision Letter · Decision Letter 1]

Dear Dr Zhang,

Thank you for your patience while your manuscript "Gnb5 ameliorates cognitive deficits by reducing β-amyloid deposition through BACE1 in Alzheimer's disease" was peer-reviewed at PLOS Biology. It has now been evaluated by the PLOS Biology editors, an Academic Editor with relevant expertise (who provided comments as Reviewer 1), and by several independent reviewers.

In light of the reviews, which you will find at the end of this email, we would like to invite you to revise the work to thoroughly address the reviewers' reports.

As you will see below, the reviewers find the study potentially interesting, but they have raised a number of important concerns. The reviewers raise concerns with data presentation, statistical rigor, and mechanistic clarity and identify issues with the western blot analyses, behavioral data, and quantifications. We think these points will need to be thoroughly addressed before we can consider your paper further for publication. I note that reviewer 3 has also raised concerns about the mouse model used here and suggests that a knock in mouse model would be more relevant to Alzheimer's Disease. While we think this is an important point - we do not think that additional mouse models are needed here to validate the findings. Instead we think that you should include explicit discussion of the limitations of the 5xFAD model. As a last point, we would like to emphasize the need to clarify the western blot data related to BACE1. As BACE1 can appear in both mature and immature forms, we think it will be important to clarify which forms are being measured in your analyses and if this has not yet been considered, you may need to add new data to flesh out these quantifications.

Given the extent of revision needed, we cannot make a decision about publication until we have seen the revised manuscript and your response to the reviewers' comments. Your revised manuscript is likely to be sent for further evaluation by all or a subset of the reviewers.

**IMPORTANT - SUBMITTING YOUR REVISION**

*Re-submission Checklist*

*Published Peer Review*

*PLOS Data Policy*

*Blot and Gel Data Policy*

Sincerely,

Luke

Lucas Smith, Ph.D.

Senior Editor

PLOS Biology

lsmith@plos.org

REVIEWS:

Reviewer #1/Academic Editor: Gnb5 is a component of G protein-coupled receptors (GPCRs) and plays a significant role in intracellular signaling. Recent GWAS studies have indicated that Gnb5 variants are associated with the onset of Alzheimer's disease (AD). However, the precise mechanism by which Gnb5 influences AD pathology remains unclear.

This study demonstrates that Gnb5 contributes to AD pathology by regulating BACE1 expression, thereby affecting Aβ generation. Specifically, Chen et al. show that overexpression of Gnb5 improves cognitive function and reduces amyloid plaque loads in AD mouse models, while Gnb5 knockdown leads to the opposite effects. The authors also identify the specific binding domains of Gnb5 (WD-40), with which it directly interacts with BACE1. In light of these findings, it would be beneficial to provide additional evidence to further clarify these results.

1. Chen et al. suggest that overexpression of Gnb5 reduces BACE1 levels, whereas knockdown of Gnb5 leads to increased BACE1 expression (Figure 5A-F). The authors are encouraged to include mRNA expression levels of BACE1 to clarify whether Gnb5 overexpression or knockdown influences BACE1 gene expression at the transcriptional level. It is required to determine whether Gnb5 affects BACE1 transcription or stability.

2. The data in Figure 6 confirm that the 28-102 amino acid region of Gnb5 is the critical site for binding to BACE1. One important aspect that warrants further exploration is the potential bidirectional relationship between Gnb5 and BACE1. Specifically, the interaction between Gnb5 and BACE1 may not only regulate BACE1 expression but also modulate Gnb5 activity and its downstream signaling. It would be valuable to explore how BACE1 binding to Gnb5 influences Gnb5 activity, and whether this modulation in turn impacts downstream signaling pathways, which could further affect BACE1 expression. Investigating this feedback loop could provide a deeper understanding of the molecular mechanisms at play in AD pathology.

3. Some of the western blot results, particularly for BACE1 (Figure 7E), require clarification. The authors should provide clearer images of the BACE1 western blots, as the current images may impact the accuracy of the quantification results.

4. Given the importance of BACE1 in AD, enzymatic activity assays would provide valuable insights into its functional role. If fluorescence-based or colorimetric assays were conducted to assess BACE1 activity, including those experimental details would strengthen the study.

5. The IBA1 immunohistochemistry images do not seem to fully align with the trends shown in the corresponding quantification graph (Figure S4). If alternative images are available that could replace the current ones, it would enhance the interpretation of Gnb5's role in the pathology.

Reviewer #2: The authors show that reduced Gnb5 levels are associated with Alzheimer's disease (AD) and that overexpression of Gnb5 in an AD mouse model leads to reduced BACE1 levels and reduced Aβ-containing plaques. They also generated a conditional Gnb5 knockout mouse to characterize its effect on cognition, and they crossed it to the AD mouse model to further modulate Gnb5 levels. The generation of these transgenic mouse models is a significant strength of the study in the investigation of Gnb5 function. The role of the single mutation in Gnb5 in the interaction with BACE1 is also intriguing, however the relationship between that finding and what has already been shown about the S81L mutation in brain function was unclear. Additional concerns are detailed below:

* The initial results suggest that pathogenic Aβ causes a decrease of Gnb5 in 6-month-old 5xFAD mice, but then the following figures focus on how changing Gnb5 levels affect Aβ levels. There is a circular effect of Aβ-Gnb5- Aβ that is not discussed in the manuscript. It could essentially be a chicken and egg problem in terms of what is the most important signaling mechanism involved.

* A neuronal marker should be used to establish whether or not the change in Gnb5 shown in Figure 1K is happening in neurons. The quantification and statistical analysis performed in Figure 1L should be done comparing each mouse per group, not a comparison across multiple individual images taken from each mouse.

* The western analyses of Gnb5 in Figure S2 are not convincing. There was no information on how the different cell types were cultured to make them pure neurons or astrocytes or microglia. That information should be added to the methods and pointed out in the results. Immunocytochemistry in cultures of immunohistochemistry in brain sections (as in Figure 1K) should be used with cell-specific markers to confirm these findings.

* It is important to explicitly acknowledge in the results section for Figure 2 that the Gnb5-CCKO mice showed a marked impairment in spatial learning in Figure 1F. This is consistent with their poor performance in the spatial memory test (Figure 2G-I) because the mice did not learn the target location during training. The authors do not point out this important distinction between spatial learning and spatial memory.

* Freezing measurements during conditioning in Day 1 should be shown including baseline freezing for experiments in Figures 2 and 3. To establish that the effect is due to memory, it needs to be established whether the mice have freezing differences during the training period.

* The green and purple groups in Figure 3F have a star suggesting a statistically significant difference, but this is not convincing because they look similar and no information on statistical analyses were provided. In Figure 3G, the average % time that control mice spent in the target quadrant is about 20% which is lower than the chance rate of 25% for each target. Thus, the control mice did not have a preference for the target quandrant, and consequently, this is not an acceptable measurement of spatial memory in this experiment. Since the controls did not show intact spatial memory as measured by % time in the target quadrant, it is then impossible to evaluate memory performance in the AD mice using the data in Figure 3G.

* Figure S3 and S4 are missing from the results. They should be described in the results, not in the Discussion. The SNAP25 in Figure 3 is way too saturated for reliable quantification. It looks like Gnb5 caused an abnormal reduction of the GluN1 subunit of NMDA receptors and Syntaxin compared to control mice. Thus, it should not be concluded that the "…overexpression of Gnb5 in AD mice mitigated the abnormal expression of these synaptic proteins".

* The tubulin blot in Figure 5A is markedly uneven which is concerning because it was used for normalization in the quantification.

* A major issue that needs to be addressed in this manuscript is that the type of statistical analyses used (for example one-way ANOVA or t-test) needs to be added to the figure legends for every individual graph that shows statistics.

* It is unclear what the following sentence means, and it needs to be clarified with more information. "Interestingly, the cognitive function related mutation site (p.Ser81Leu (S81L)) on Gnb5 also locates in this region[18, 19]"

* The groups in Figure 7C and D are not explicitly labeled. The graphs need a legend. Mice should be labeled as 5xFAD.

* It is also unclear what the sentence means in lines 253-255 of the Discussion. Indeed, there is limited success for that strategy. Wouldn't this suggest that this strategy should not be developed as treatments for AD? In general, the Discussion section could include more information relating the findings to previous literature instead of restating results.

* The claim that "with heterozygosity of Gnb5 shown to exacerbate… in AD model mice" is incorrect. The two studies cited are human subjects studies, which show correlative effects. They did not show that Gnb5 exacerbates disease.

* The NOR test was in the methods but not in the paper.

* There are many grammatical errors throughout the manuscript including Lines 81-83,101, 146-148, 181-182, 293-297, Figure 7C and D y-axes. There were many more grammatical errors in the methods section which is poorly written with important missing information.

* Minor: One of the images in Figure 4A is flipped in the opposite orientation.

Reviewer #3: This is in many ways an interesting paper but it leaves the reader with almost more questions than answers. There are many G-Protein-mediated responses in which beta subunits are involved. It is not clear whether the authors are suggesting that all the changes suggested in this paper to be caused by loss of or overexpression of this particular beta subunit are mediated via the same initial ligands/receptors. Alternatively, as changing the level of this beta subunit will affect many different pathways it could have a wide range of independent effects. Whether changing the level of this beta subunits sets off a chain of reactions related to BACE activity or a series of different unconnected effects is unclear. No doubt loss of or overexpression of this subunit may also affect many other phenomena.

In Alzheimer's disease there is a loss of synapses within and in the immediate vicinity of plaques and so one of the proteins or local expression systems affected may well be this beta subunit. It is notable that various alpha and at least one gamma subunit are also down. This is illustrated in Fig 1 K-L in which measuring the fluorescence in a field covered by a plaque in the 5xFAD mouse almost wipes out any Gnb5 as it may will be lost with the loss of synapses in that region whereas, whole tissue analysis with Western blots is less extreme. Indeed in the left hand panel of 1K the holes are visible where there are plaques but the background levels are not so different between WT and 5xFAD. Note that for the statistics shown in Fig 1L this is also pseudo replication. The sample size should be n=4 with separate observations from the same mouse averaged to give a single data point. Fig 1 suggests that the main beta subunits in the G-proteins of lost synapses is probably Gnb5 and that likely it is expressed locally within synapses.

The observation that altering various G-protein-mediated pathways by altering the level of available subunits affects behaviour, is not surprising but there is little evidence here that this has anything to do with BACE or Alzheimer's disease. That some of these behaviours occur in the 5xFAD mice is probably unrelated to Alzheimer's disease. Like several overexpression models 5xFAD show various behavioural deficits but this is most likely a factor of overexpression of APP and may largely relate to the alpha side of the APP pathways. Note that the statement starting line 172 "The accumulation of Aβ in the brain is a hallmark of AD and is strongly associated with cognitive decline in patients ....." is clearly incorrect. Plaques are not well correlated with cognitive decline and indeed in humans diagnosis only occurs once there is a substantial load of neurofibrillary tangles. Hence substantial cognitive deficits in mice with plaques but no tangles is unlikely to be related to the cognitive deficits of Alzheimer's disease. Overall, now that knock in technology is a decade old with mouse knock in models readily available, the ongoing use of transgenic overexpressing mice is rather unfortunate and limiting.

In terms of the plaque data in Figure 4. Firstly this is again a classic case of pseudoreplication. The sample size is 3, not around 18 as illustrated here. The results from the sections from the same mouse should be averaged to give a single value for each mouse. It is impossible to tell here how consistent the results within any mouse are. For example, the lowest values in the purple column in Fig 4B could all come from one mouse and the highes values in the green column from another with the remaining 4 mice (2 from each group having similar overlapping values). Similarly in figs 4F&G it is very likely that, particularly in the blue columns that the data fall into the 3 groups of the individual mice with one mouse tending to skew the data to a higher mean.

It is interesting that a beta subunit of a G-protein coimmunoprecipitates with BACE1. In this case Figs 5 and 6 the HEK cell experiment seem to suggest a direct binding of a particular domain of this G-protein subunit although how, in a physiological system, this would then relate to the activation of the G-protein by a set ligand or just a direct effect of this protein is unclear. Is the Ser81 of the beta subunit essential for the normal activation of the G-protein in the observations in Fig 7 or is this an effect related to the binding to BACE observed in HEK cells? Note the problem of pseudoreplication is again present in Fig 7 C&D.

Overall there are some interesting findings in this study but it is difficult to interpret the significance of these findings or the internal relationship between them, without relating the effects of altering a particular beta subunit to the specific pathways that mediate the changes seen

---

## [Decision Letter · Decision Letter 2]

Dear Dr Zhang,

Thank you for your patience while we considered your revised manuscript "Gnb5 ameliorates cognitive deficits by reducing β-amyloid deposition through BACE1 in Alzheimer's disease" for publication as a Research Article at PLOS Biology. This revised version of your manuscript has been evaluated by the PLOS Biology editors, the Academic Editor and the original reviewers.

The reviews are appended below, and you will see that all three reviewers think the paper has been strengthened in the revision. However, reviewer 3 highlights that the study has not conclusively shown that the effects of Gnb5 on cognition are mediated by its modulation of BACE1, and we think this is an important point that should be addressed with textual changes.

Based on the reviews, we would like to invite you to revise your study, to address reviewer 3's last concerns. We think that it would still be OK for you to lay out the case for why you think the effects of Gnb5 on cognition are likely mediated by changes in BACE1 in the discussion section, but we think that you should also explicitly acknowledge reviewer 3's point in the limitations section, and include a more detailed discussion of potential alternative mechanisms, such as classical G-protein signaling pathways, that might contribute to the cognitive effects observed. We also think that the claims in the title and abstract will need to be toned down. For example, to address reviewer 3's concerns, we think the title could be changed to:

"Gnb5 is a negative regulator of the BACE1-mediated Aβ generation and ameliorates cognitive deficits in a mouse model of Alzheimer's disease"

We expect to receive your revised manuscript within two weeks.

*Published Peer Review History*

*Press*

Sincerely,

Luke

Lucas Smith, Ph.D.

Senior Editor

lsmith@plos.org

PLOS Biology

Reviewer remarks:

Comments from the Academic Editor (who was originally Reviewer #1): Most of my concerns have been addressed adequately.

Reviewer #2: The authors have adequately addressed the concerns.

Reviewer #3: In most cases the changes made and responses from the authors address my concerns well. The one point that I think still needs clarification is the assumption that the behavioural effects seen are related to the BACE1 modulation or indeed to Alzheimer's disease. There is a logical disconnect in the authors response.

1. It is true that they have demonstrated that the overexpression of GnB5 reduces amyloid beta production through BACE1 and that this is probably through an non canonical pathway rather than classic g-protein modulation. Moreover overexpression of GnB5 decreases amyloid load.

2. They have also demonstrated that there is a decrease in GnB5 in the 5xFAD mice and that knocking out GnB5 causes behavioural deficits while overexpression of GnB5 in 5xFAD mice rescues the behavioural deficits.

However they have not demonstrated that the behavioural changes are due to the BACE1 effects, rather than due to some other classical G-protein mediated pathway. The behavioural changes seem to be dependent on GnB5 levels but this would affect many pathways and so it cannot be assumed that just because BACE1 is one of the pathways affected that the behavioural changes are due to modulation of BACE1.

This needs to be clarified in the discussion and the title should be changed accordingly

---

## [Editor Report · Decision Letter 3]

Dear Dr Zhang,

Thank you for the submission of your revised Research Article "Gnb5 is a negative regulator of the BACE1-mediated Aβ generation and ameliorates cognitive deficits in a mouse model of Alzheimer's disease" for publication in PLOS Biology and thank you for addressing the last reviewer requests in this revision. On behalf of my colleagues and the Academic Editor, Dong-Gyu Jo, I am pleased to say that we can in principle accept your manuscript for publication, provided you address any remaining formatting and reporting issues. These will be detailed in an email you should receive within 2-3 business days from our colleagues in the journal operations team; no action is required from you until then. Please note that we will not be able to formally accept your manuscript and schedule it for publication until you have completed any requested changes.

PRESS

We frequently collaborate with press offices. If your institution or institutions have a press office, please notify them about your upcoming paper at this point, to enable them to help maximize its impact. If the press office is planning to promote your findings, we would be grateful if they could coordinate with biologypress@plos.org. If you have previously opted in to the early version process, we ask that you notify us immediately of any press plans so that we may opt out on your behalf.

Sincerely, 

Luke

Lucas Smith, Ph.D.

Senior Editor

PLOS Biology

lsmith@plos.org